

# A new aggregation and riming discrimination algorithm based on polarimetric weather radars

Armin Blanke[1], Mathias Gergely[2], and Silke Trömel[1,3]

[1]Institute of Geosciences, Department of Meteorology, University of Bonn, Bonn, 53121, Germany
[2]German Meteorological Service (Deutscher Wetterdienst, DWD), Observatorium Hohenpeißenberg, Hohenpeißenberg, 82383, Germany
[3]Laboratory for Clouds and Precipitation Exploration, Geoverbund ABC/J, Bonn, 53121, Germany

**Correspondence:** Armin Blanke (ablanke1@uni-bonn.de)

**Abstract.** The distinction between riming and aggregation is of high relevance for model microphysics, data assimilation and warnings of potential aircraft hazards due to the link between riming and updrafts and the presence of supercooled liquid water in the atmosphere. Even though the polarimetric fingerprints for aggregation and riming are similar qualitatively, we hypothesize that it is feasible to implement an area-wide discrimination algorithm based on national polarimetric weather radar networks only. Quasi-vertical profiles (QVPs) of reflectivity ($Z_H$), differential reflectivity ($Z_{DR}$) and estimated depolarization ratio (DR) are utilized to learn about the information content of each individual polarimetric variable and their combinations for riming detection. High-resolution Doppler spectra from the vertical (birdbath) scans of the C-band radar network of the German Meteorological Service serve as input and ground-truth for algorithm development. Mean isolated spectra profiles (MISPs) of the Doppler velocity are used to infer regions with frozen hydrometeors falling faster than $1.5\ \mathrm{ms}^{-1}$ and accordingly associated with significant riming. Several machine learning methods have been tested to detect riming from the corresponding QVPs of polarimetric variables. The best performing algorithm is a fine-tuned gradient boosting model based on decision trees. The precipitation event on 14 July 2021, which led to a catastrophic flooding in the Ahr valley in western Germany, was selected to validate the performance. Considering balanced accuracy, the algorithm is able to correctly predict 74 % of the observed riming features and thus, the feasibility of reliable riming detection with national radar networks has been successfully demonstrated.





## 1 Introduction

The reliable detection and prediction (classification) of riming based on ground-based remote sensing observations is a crucial but not trivial endeavor. Despite the wealth of information offered by polarimetric radar measurements and their numerous associated advances in research, the distinction between dominant aggregation and riming processes using weather radar has been questioned to date. In this study, machine learning is exploited to reveal the relationship between different polarimetric variables and their combinations and dominating riming processes in radar-monitored precipitation cells.

Ice crystals are subjected to a variety of microphysical processes during their lifetime as they fall to the ground (e.g., Kumjian et al., 2022). The most fundamental growth processes in the ice phase are aggregation, riming and vapor deposition. However, these processes also occur in combination and with fast transitions within the evolution of ice phase particles (DeLaFrance et al., 2024). During aggregation processes two or more ice crystals stick together through ice-ice collisions (Field et al., 2017) to form a single larger particle, transforming dense individual ice crystals into aggregated particles with reduced density but similar water content. In contrast to aggregation, riming describes the process when an ice particle collects supercooled liquid cloud droplets (ranging in size from microns to tens of microns), thus, ice water content increases at the expense of liquid drops. These rimed particles typically exhibit enhanced fall velocities (Kumjian et al., 2016) due to their rapid increase in mass and density. They can reach large sizes and become more isotropic (e.g., Maahn et al., 2024). While the increase in size and thus reflectivity ($Z_H$), as well as the decrease in differential reflectivity ($Z_{DR}$) is evident in both aggregation and riming, densely rimed particles fall with up to twice the speed of an equivalent unrimed particle with exactly the same maximum dimension (Locatelli and Hobbs, 1974).

The distinction between aggregation and riming below the dendritic growth layer (DGL; Ryzhkov and Zrnić, 2019), usually located between -10 and -15°C, is important as the latter signals the presence of supercooled liquid water (SLW) and its accretion to the airframe (Serke et al., 2011) and critical flight data sensors (Milani et al., 2024) may produce an in-flight icing hazard for aircraft (Ellis et al., 2012). These dangerous conditions can be observed before or during the presence of riming signatures as long as SLW is not fully depleted. Overall, riming represents a key process as a large percentage of cloud systems contain SLW (Hogan et al., 2003), especially below the DGL. SLW may also trigger additional ice growth via the Wegener–Bergeron–Findeisen process (Wegener, 1912; Bergeron, 1935; Findeisen, 1938). Furthermore, riming favors secondary ice production through the Hallett-Mossop ice multiplication process, also known as rime splintering (Hallett and Mossop, 1974), which is active between -3 and -8°C. Thus, future benefits and applications of the envisioned area-wide riming detection algorithm based on slant-viewing polarimetric weather surveillance radars only are manifold. It supports and improves process understanding and enables detailed model evaluation. Also, state-of-the art polarimetric microphysical retrievals (e.g. of ice water content, number concentrations and mean volume diameters) show convincing accuracy and encourage their use for model evaluation and data assimilation (Blanke et al., 2023). However, these retrievals are not designed for riming conditions when graupel or even hail can be present. With a riming detection on hand, the assimilation of such modified retrievals into numerical weather prediction models could be restricted to regions where enhanced accuracy can be expected to further improve e.g. quantitative precipitation forecasts.





Since cloud droplets show mostly less than 50 $\mu m$ in diameter, the direct detection of SLW with weather radars is not possible.
Instead, past studies repeatedly employed the mean Doppler velocity from profiling radars to detect and study riming. E.g.,
Mosimann (1995) derived a quantitative relationship between radar Doppler velocities of a vertically pointing X-band radar
and riming in stratiform precipitation. Fall velocities of unrimed snow particles do not exceed $\sim 1$ ms$^{-1}$ (Locatelli and Hobbs,
1974; Karrer et al., 2020), because during the aggregation process the impact of increasing mass on the terminal velocity is to a
great extent balanced out by the additional air drag (Zawadzki et al., 2001). However, substantially rimed particles can exhibit
fall velocities ranging from 1.5 to 2.5 ms$^{-1}$ or even faster (e.g., Vogel et al., 2015; Matrosov, 2023).

To set-up the area-wide riming detection algorithm based on slant-viewing polarimetric weather surveillance radars only, we
utilized and analyzed quasi-vertical profiles (QVPs; Trömel et al., 2013, Ryzhkov et al., 2016) of $Z_H$, $Z_{DR}$ and in particular
depolarization ratio (DR). Ryzhkov et al. (2017) introduced DR as a good proxy for radar circular depolarization ratios and a
potential candidate for the detection of riming. Due to the inherent noise reduction and the presentation of polarimetric vari-
ables in a time versus height format, QVPs facilitate the detection of fingerprints for dominating microphysical processes and
their temporal evolution in a sufficiently homogeneous cone spanned above the radar. The polarimetric fingerprints for (heavy)
riming and aggregation are qualitatively the same, exhibiting an increase in $Z_H$ and decreases in $Z_{DR}$ and specific differential
phase $K_{DP}$, unless a substantial concentration of columnar ice crystals is simultaneously prevalent, which may lead to an
observable increase instead of decrease in $K_{DP}$ (e.g., Kumjian, 2012; Kumjian et al., 2022). However, the time-height format
of QVPs enables the investigation and quantification of the relationships between different polarimetric variables and Doppler
velocities. In this study, profiles of Doppler spectra, which can be interpreted as a distribution of particle fall velocities super-
imposed with vertical air movements as a function of height (Fabry, 2015), are used to introduce and train a radar algorithm
for the discrimination between the two processes. Similar to $Z_{DR}$, DR is lower in rimed snow than in aggregated snow, but the
corresponding difference in DR is generally larger. While $Z_{DR}$ differs by 0.2–0.4 dB, DR differs by 2–4 dB between these two
processes (e.g., Ryzhkov et al., 2017). Such differences are clearly evident in QVPs.

A variety of techniques resp. machine learning methods for classification are available in the literature. In this study, we fo-
cused on four approaches, namely logistic regression (LR; Wilks, 2011), a quadratic discriminant analysis (QDA; Geisser,
1964), gradient boosting machine (GBM; Friedman, 2001) and multilayer perceptron (MLP) artificial neural network to set-up
the algorithm. One great advantage of these methods lies in their ability to sift through large amounts of training data and
discover meaningful patterns that are not easily discernible to humans.

The article is structured as follows: In Sect. 2 an overview of the remote sensing observational data base and processing tech-
niques is provided. Sect. 3 introduces the different methods tested as well as the performance metrics considered throughout
this work, while Sect. 4 details DR and the algorithm development. The main results and verification are presented in Sect. 5.
The final algorithm is subsequently applied to an independent riming case, followed by an elaboration on the main advantages
along with the limitations of the newly proposed algorithm. Sect. 6 closes with a summary and a comprehensive discussion of
directions for future research and refinement opportunities.



## 2   Data and processing

The quality of the training data is key for the performance of algorithms developed with machine learning techniques. Schultz et al. (2021) pointed out that the proper selection and preparation of data is of crucial importance in order to achieve good and generalizable results. Accordingly, this section presents the preparation and processing of the radar data used.

### 2.1   Polarimetric C-band radar data

Our analysis is based on observations of DWD's national C-band (wavelength ≈ 5.6 cm) weather radar network including 17 state-of-the-art polarimetric Doppler radars continuously performing 3-D volume scans in a 5 min scan schedule. These include plan position indicator scans (PPIs) measured at 10 radar elevation angles between 0.5 and 25 degrees, each with a resolution of one degree in azimuth and 0.25 km in range. Typical maximum slant ranges are about 180 km. At higher elevation angles of more than 8 degrees, the maximum slant range decreases to around 60 km. A vertically pointing, so-called birdbath scan, ends the five minute sampling sequence. More detailed information of the scanning routine, radar systems and data processing at DWD can be found in Helmert et al. (2014) and Frech et al. (2017).

This study explores riming cases observed at DWD's Essen radar site (ESS; Fig. 1) located in western Germany in order to

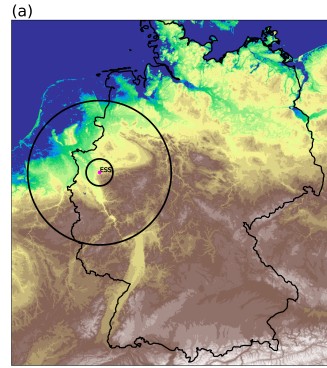
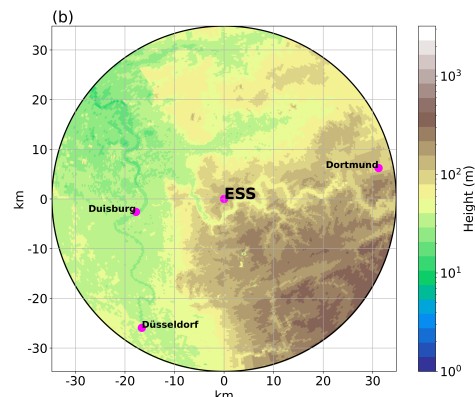

**Figure 1.** Panel **(a)** shows the geographic location (magenta dot) and area covered by the operational Essen radar (ESS; lat: 51.405649°N, lon: 6.967111°E; alt: 185.11 m a.s.l.) in western Germany with the PPI at one degree elevation angle. The larger circle indicates the approximate maximum range of 180 km around the radar and the smaller circle the coverage limited to a range of 35 km. Panel **(b)** displays a zoom into the limited ESS region. Colors indicate the terrain height of the study area in m a.s.l. . A selection of surrounding cities are also indicated with magenta dots.

train the riming algorithm. These data include five stratiform precipitation events monitored on 13 May 2021, 24 July 2021, 3 November 2021, and two time segments on the 2 January 2022. Furthermore, one additional event on 14 July 2021 is used for final evaluation (Table 1).

QVPs of polarimetric variables are generated based on PPIs measured at 12 degree elevation, enabling the joint analysis with the birdbath data. We constrained the calculation of the QVPs to a maximum range of 35 km (Fig. 1) in order to on the one hand





**Table 1.** List of events including the time periods from consecutive birdbath scans recorded at ESS radar and their utilization purpose. While the event on 14 July 2021 (Ahrtal flooding) is used as independent data set for the evaluation of the riming algorithm, the remaining events are used for algorithm development and referred to as initial data set in Sect. 4 and Fig. 5.

| Time periods, dates | No. of riming periods | Note |
|---|---|---|
| 18:00 - 20:30 UTC, 13 May 2021 | 1 | development |
| 14:00 - 19:00 UTC, 14 July 2021 | 1 | evaluation |
| 13:00 - 20:00 UTC, 24 July 2021 | 1 | development |
| 11:30 - 15:00 UTC, 03 November 2021 | 1 | development |
| 03:00 - 09:30 UTC and 17:30 - 21:00 UTC, 02 January 2022 | 2 | development |

improve the comparability and on the other hand still cover sufficiently high altitudes. Preceding quality control, calibration and preprocessing of the radar data is performed as follows:

To mitigate the impact of noise and non-meteorological scatterers, data is filtered with a cross-correlation coefficient $\rho_{hv} \geq 0.8$. Noise corrections following Ryzhkov and Zrnić (2019) are applied to $\rho_{hv}$ and the theoretical $Z_H$-$Z_{DR}$ relationship for C-band in light rain (Ryzhkov and Zrnić, 2019) is used to calibrate $Z_{DR}$. Due to the identified elevation dependency of the

offset, this calibration method was preferred to the use of the birdbath scan. Furthermore, only radar data with a signal to noise ratio (SNR) greater than 10 dB are taken into account after the correction of $\rho_{hv}$. For the development of the riming algorithm, it is particularly important to exclude events with pronounced up- and downdrafts associated with convection. Thus, only stratiform precipitation with a detectable melting layer (ML) is considered. So, after QVP calculation the ML detection strategy introduced by Wolfensberger et al. (2016) is used to derive a first-guess estimate of the ML locations and then adjusted

to nearby locations where $\rho_{hv}$ returns to values above 0.97 (Giangrande et al., 2008). The precise detection allows not only the selection of stratiform rain, it also enables to restrict to the pure ice phase above the ML.

## 2.2 Doppler spectra

Doppler spectra can provide high-resolution profiles of radar equivalent reflectivity factor, mean Doppler velocity (MDV), and spectrum width, and are widely used to determine microphysical and dynamical properties of clouds (e.g., Kollias et al.,

2007; Kalesse et al., 2016; Von Terzi et al., 2022; Billault-Roux et al., 2023). Only since the update of the scan schedule for the national radar network of the DWD on 18 May 2021 are Doppler spectra stored for the entire C-band radar network on a regular basis. Previously, the operational birdbath scan was mainly used for the calibration of $Z_{DR}$ (Frech and Hubbert, 2020) but the Doppler spectra now provide new opportunities also for operational applications. This study employs them as ground-truth for riming occurrences.

The flexible multistep post processing of the Doppler spectra as described in Gergely et al. (2022) is performed to isolate the weather signal from non-meteorological echoes exploiting polarimetric attributes (e.g. the signal power in one of the two available polarization channels, the absolute value of the uncalibrated spectral differential reflectivity $sZ_{DR}$ and the texture of $sZ_{DR}$), to calculate the properties of each precipitation mode identified, and potentially also multimodal characteristics, if more





than one mode is present. These characteristics, such as bimodal amplitude and separation (as defined in Zhang et al., 2003),

are used to quantify the relation among the individual, simultaneously occurring precipitation modes. Figure 2 demonstrates

the performance of the method. The complete spectra (left panel) include unwanted non-meteorological contributions and static

clutter at Doppler velocities close to 0 ms$^{-1}$ at all heights. These artifacts are removed in the processed spectra (right panel)

together with the antenna near-field, which extends up to a height of about 650 m at all Doppler velocities and determines the

minimum valid height. The weather signal reaches up to an altitude of about 8 km with a transition from frozen precipitation to

much faster falling rain at heights about 2 km above the radar. Rain below the ML shows a broader distribution and also higher

fall velocities of up to -6 ms$^{-1}$. Moreover, this isolated spectrum shows evidence of significantly rimed snow above the ML.

It contains two precipitation modes between approximately 2 km and 3 km height. The primary mode is characterized by fall

velocities larger than -2 ms$^{-1}$, typical for rimed particles, while the second mode exhibits reduced velocities around -1 ms$^{-1}$.

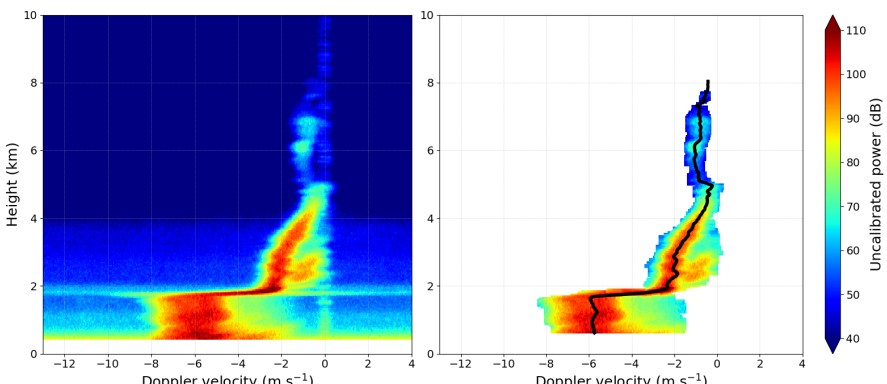

**Figure 2.** Mean Doppler power spectra of an exemplary 15 s birdbath scan recorded on 2 January 2022 at 03:40 UTC. The direct output (left), after the internal radar signal processor had already applied a notch filter to mitigate strong clutter near 0 ms$^{-1}$ for each individual Doppler spectrum, is shown together with the isolated average Doppler spectra after postprocessing (right). Colors indicate the uncalibrated radar signal power (in dB). The black line indicates the corresponding mean profile of power-weighted mean velocity.

So far, Doppler spectra recorded with DWD's C-band radars have only been used to study the profiles of individual birdbath

scans in detail (Trömel et al., 2021; Gergely et al., 2022). By applying in the ensuing step the novel mean isolated spectra

profile (MISP) technique, time series of the processed spectral data can now also be displayed in a convenient time vs. height

format, allowing a direct comparison with polarimetric QVPs. The MISP technique uses the mean of the isolated spectra

(e.g, right panel in Fig. 2) at each height level with all included precipitation modes. Note that, the derived MDV from the

isolated spectra contains a weighting and is therefore calculated from the power-weighted mean velocity $\bar{v}$ (in ms$^{-1}$) for all

precipitation modes, with the Doppler power in each individual spectral bin denoted by $S$, thus explicitly accounting for the

spectral dependence on Doppler velocity $v$:

$$\bar{v} = \frac{\sum_i v_i S(v)}{\sum_i S(v_i)}. \tag{1}$$



The resulting mean profiles in the time versus height format are then referred to as MISP. The MISPs of MDV allow the distinction between various hydrometeor types. Note, only stratiform precipitation events should be considered to minimize
misinterpretation due to vertical air motions. Temporal averaging over each 15 s birdbath scan already dampens the effects due to large-scale vertical air motion and reduces the influence of turbulence on the measured Doppler velocities of the falling precipitation particles.

As an example, Fig. 3 shows MISPs of mean power (in dB) and MDV for a riming event on 2 January 2022. The rapid increase

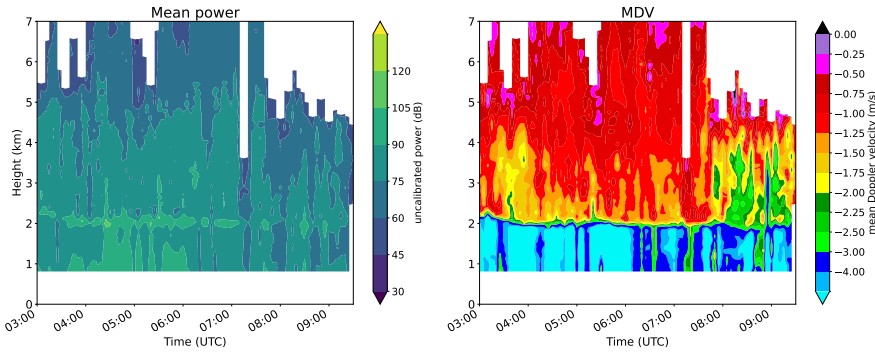

**Figure 3.** MISPs of mean power (left) and MDV (right) observed by ESS radar recorded on 2 January 2022 between 03:00 and 09:30 UTC. Negative MDV values indicate motion towards the radar.

in Doppler velocities at an altitude of around 2 km indicates the transition from the ice to the liquid phase and is in agreement
with the detected ML height in the QVPs (not shown). Above the ML, the event shows MDVs exceeding 1.5 ms$^{-1}$, clearly indicating riming processes (Kneifel and Moisseev, 2020). However, the potential occurrence of densely rimed dendrites, and lightly rimed aggregates, e.g. during the fill-in stage of riming growth (Heymsfield, 1982), is challenging to detect with such a fixed threshold value of 1.5 ms$^{-1}$. The fall velocity of such rimed particles may overlap with the ones of unrimed aggregates. Decreasing air density with height impacts the fall velocity of hydrometeors and needs to be taken into account to avoid
misinterpretation. Therefore, raw MDV$_r$ are transformed into fall velocities at surface conditions. Following Heymsfield et al. (2013), the pressure $p$- transformed MDV at altitude $z$ is given by

$$\text{MDV}(z) = \text{MDV}_r(z) \left[\frac{p(z)}{p_{ref}}\right]^{0.4}, \tag{2}$$

where $p_{ref}$ is the reference pressure at the surface. In this study profiles $p(z)$ were obtained from radio soundings of the worldwide repository hosted at the University of Wyoming (http://weather.uwyo.edu/upperair/sounding.html, last access: 27
March 2024) for the permanent sounding station Essen (station number 10410), which is the only directly collocated sounding station available for the DWD network. In the following the transformed MDV$_r$ is referred to as MDV and interpreted as the typical particle fall velocity. Finally, the rime mass fraction (RMF), defined as the fraction of total particle mass obtained by riming (Kneifel and Moisseev, 2020), can be derived from the MISPs of MDV.





# 3 Methodology

First Sect. 3.1 presents a general description of the learning techniques utilized, followed by an overview of all statistical metrics used for evaluation in Sect. 3.2.

## 3.1 Description of the learning techniques

Several approaches to classify rainy observation periods as dominated by riming processes or not based on polarimetric radar variables only were tested. Besides a simple threshold-based approach (TB), the relationship between Doppler velocities and different polarimetric variables can also be learned (e.g. by a supervised neural network). Therefore, we investigated four methodologies: an LR, a QDA, a GBM based on decision trees and an MLP ANN trained with a commonly used back propagation algorithm (Table 2). In the following, the basic principles of the selected methods are briefly described together with the respective hyperparameters that define the architecture and training process if needed. Tools from the *scikit-learn* Python machine learning library (Pedregosa et al., 2011) were utilized for all the learning methods. For more detailed information on these methods and their mathematical formulations we refer to the *scikit-learn* user guide.

**Table 2.** Overview of classification techniques.

| Classifier | Abbreviation | Relationship | Citation |
| --- | --- | --- | --- |
| threshold-based approach | TB | linear | |
| logistic regression | LR | linear | Wilks (2011) |
| quadratic discriminant analysis | QDA | non-linear | Geisser (1964) |
| gradient boosting machine | GBM | linear and non-linear | Friedman (2001) |
| artificial neural network | ANN | linear and non-linear | Schmidhuber (2015) |

The most basic detection method for riming, that defines the TB herein, relies on threshold criteria of the polarimetric variables estimated via scatterplots of the variables relative to the MDVs faster than $1.5\,\mathrm{ms^{-1}}$. Expectations with respect to riming based on prior studies (e.g., Ryzhkov et al., 2016) are also taken into account for the selection of these thresholds.

As a first statistical analysis method, LR is used to model the dependence of a binary response variable on one or more explanatory variables. The probability of an event success is modeled by taking the log-likelihood for the event to be a linear combination of one or more independent variables. LR is easy to implement, interpret and efficient to train.

The QDA algorithm is a classic and flexible classifier with a quadratic decision surface, that minimizes the total probability of misclassification and allows for non-linear separation of data. Therefore, it fits Gaussian density (covariance matrix) to each class. No assumption on identical covariances for each class is required. After modeling the likelihood of the classes with a supervised method, the QDA uses a normal distribution to make predictions. A QDA algorithm is easy to compute (no hyperparameters to tune) and inherently multiclass.

The GBM is a learning technique based on decision trees (Breiman, 1996). It is a generalization of tree-boosting in which the





learning task is posed as a numerical optimization problem. Boosted trees are comparable to random forests (Breiman, 2001) in the sense that an ensemble of decision (regression) trees are considered and calculated. As opposed to bagging during the ensemble and resampling processes, a boosting procedure is considered as a technique in which simple parameterized models are sequentially added to the ensemble at each iteration.

At the beginning, the number of maximum splits within each tree (also known as the depth) is specified as a hyperparameter in the process of model tuning. In order to reduce variance and bias, each tree is computed as a function of its predecessors and weighted according to its accuracy. The gradient descent procedure is used to iteratively update the weights and minimizes the difference from the function predicting the actual observation. Using numerous model outputs in combination is advantageous to further reduce biases.

In general, an ANN represents a mathematical model trained to recognize patterns and to make predictions. A MLP is a fully connected neural network that consists of an input layer, several hidden layers (artificial neurons) as well as an output layer. It typically performs a sequence of matrix multiplications, followed by an element-wise non-linear function (the activation function) for each iteration. These allow the network to learn linear and non-linear relationships. Similar to the GBM, hyperparameters such as the number of hidden layers (referred to as the layer depth), the corresponding number of neurons, the type of activation function and the initial learning rate need to be tuned to derive the optimal architecture for the ANN. In this study, a fairly simple ANN structure can be employed, which generally reduces the risk of overfitting and requires less computation.

## 3.2 Scores and performance metrics

The performances of the proposed riming retrievals are evaluated by computing multiple pertinent scores to ensure robustness of the evaluation procedure. Here, we convert the ground truth and the results of the retrievals into binary fields (riming yes/no), in other words, presence of riming or lack thereof, in order to simplify the analysis and efficiently apply and adapt it to our needs.

Four distinct types of metrics, including true negatives (TN), false positives (FP), false negatives (FN) and true positives (TP), are broadly used to assess the performance of binary classification analyses. Based on those, the accuracy (ACC), precision (PR), true negative rate (TNR; referred to as specificity), and recall (RC; also known as sensitivity in diagnostic binary classification) are defined as follows:

$$ACC = \frac{TN + TP}{TN + TP + FN + FP}, \tag{3}$$

$$PR = \frac{TP}{TP + FP}, \tag{4}$$

$$TNR = \frac{TN}{TN + FP}, \text{ and} \tag{5}$$





$$RC = \frac{TP}{TP + FN}.$$

(6)

The distinct metrics can be displayed in a 2×2 contingency table (Pearson, 1904), which is also referred to as the so-called

confusion matrix (Miller and Nicely, 1955)

$$\boldsymbol{M} = \begin{pmatrix} TP & FN \\ FP & TN \end{pmatrix}$$

(7)

summarizing the results of the classification. The standard ACC ranges in the real unit interval [0,1]. The highest possible value

of 1 corresponds to perfect classification, whereas 0 is the lowest possible value indicating clear misclassification.

Overall, ACC tends to provide a too optimistic assessment of the classification ability if the category to be detected is underrep-

resented, i.e. ACC is not adequate to quantify the performance of an unbalanced data set. In general, an evaluation metric alone

is only able to reflect part of the model's performance (Wang et al., 2024). One alternative is the commonly used balanced

ACC (BA), which is the arithmetic mean of sensitivity RC and specificity TNR:

$$BA = \frac{RC + TNR}{2}.$$

(8)

Again, BA ranges between 0 and 1 and is an appropriate metric dealing with unbalanced data sets.

Further, the F1 score represents a harmonic mean of PR and RC and is calculated as

$$\text{F1 score} = 2 \cdot \frac{PR \times RC}{PR + RC} = \frac{2 \cdot TP}{2 \cdot TP + FP + FN},$$

(9)

reaching the value 0 in case of clear misclassification and the best value 1 for perfect classification. This metric is more sensitive

to changes in the detection of positives, because, unlike ACC, the F1 score does not take into account TN (not symmetric).

Excluding TN can be especially beneficial because it can dominate classification tasks in meteorology due to the often rare

nature of events (Chase et al., 2022), e.g. the occurrence of riming.

Matthews correlation coefficient (MCC; Matthews, 1975) is another measure for the quality of binary classifications, which is

not affected by imbalanced data sets:

$$\text{MCC} = \frac{TP \times TN - FP \times FN}{\sqrt{(TP + FP)(TP + FN)(TN + FP)(TN + FN)}}.$$

(10)

MCC, also known as phi coefficient, is a correlation coefficient value ranging between -1 and +1. A coefficient of +1 represents

a perfect prediction, 0 an average random prediction and -1 an inverse prediction. However, MCC represents a binary classifier

that yields a high score only if the binary predictor was able to correctly predict the majority of positive and the majority of

negative outcomes. Also the normalized MCC, hereafter defined as NMCC=(MCC+1)/2, can be useful since it linearly projects

MCC onto the range interval from 0 to 1.

The Jaccard index (Jaccard, 1901), also termed as "intersection over union" (Wilks et al., 1990) and frequently referred to as the

critical success index (CSI; Donaldson et al., 1975) in meteorological literature, is a statistic used for comparing the similarity




and diversity of finite sample sets and defined as the ratio between the size of the intersection and the size of the union of the sample sets A and B:

$$J = \frac{|A \cap B|}{|A \cup B|} = \frac{TP}{TP + FP + FN} \tag{11}$$

with values ranging between 0 (no overlap) and 1 (complete overlap). Like the F1 score, $J$ does not consider TN.

Lastly, the widely used Cohen's Kappa ($\kappa$) expresses the level of agreement between two sets and takes into account the agreement occurring by chance. In meteorology it is also known as the Heidke skill score (Heidke, 1926) and is calculated via

$$\kappa = \frac{2 \times (TP \times TN - FN \times FP)}{(TP + FP) \times (FP + TN) + (TP + FN) \times (FN + TN)} \tag{12}$$

with values ranging from -1 to 1. Despite known disadvantages like e.g. the high sensitivity to the distributions of the marginal totals, it is included as one of the most popular metrics used in machine learning for comparison.

In order to quantify the role of each individual polarimetric variable as predictor in the riming algorithm, Shapley values (Shapley et al., 1953) are used as in Buschow et al. (2024). The Shapley values, originally developed in game theory, provide information on how the payout (prediction) can be fairly distributed among the predictors (also denoted as features). Note, these calculated contributions always add up to the total amount, however, the importance of a variable may vary for each performance measure considered. Essentially, the calculated Shapley values allow for a ranking of input features according to

their relevance. The inherent impurity-based feature importance (also known as Gini importance or mean decrease impurity; Breiman, 2001) is additionally used, as it indicates the importance within the same model, which does not require recalculation and tuning of hyperparameters. For random forests, it is defined as the total decrease in node impurity averaged over all trees of the ensemble and measures the amount each feature contributes to the reduction in variance of the model when that feature is used to split the data. In contrast to Shapley values, this performance measure is considered biased towards features with

high cardinality (Grömping, 2009), i.e. a large number of distinct values.

## 4 Developing a riming detection algorithm for C-band radars

Sect. 4.1 emphasizes DR as a promising proxy for ongoing riming processes and details the microphysical information content of the to date still underutilized polarimetric variable DR in general, while Sect. 4.2 describes the entire workflow of the riming detection algorithm development.

### 4.1 Depolarization ratio

The impact of riming on $Z_{DR}$ is twofold. On the one hand, the ice particles become more spherical due to riming, which leads to a reduction in $Z_{DR}$. On the other hand, the associated increase in density is supposed to increase $Z_{DR}$. Observations indicate that the impact of the particle shape dominates the impact of density, resulting in a small overall reduction of $Z_{DR}$ in case of heavy riming (Ryzhkov et al., 2016). DR, however, proved to be a useful parameter for characterizing the microphysical

properties of snow and shows a more pronounced riming fingerprint (e.g., Ryzhkov et al., 2017).





DR can be derived from measurements of dual-polarization radars operating in SHV mode (simultaneous transmission/reception of orthogonally polarized waves) and represents a good proxy for the circular depolarization ratio (CDR) measured by radars with circular polarization (Matrosov, 2004; Ryzhkov et al., 2017). Thus, DR can be derived based on measurements of the DWD network and included as predictor in the envisioned riming algorithm. In case of dry aggregated snow with low bulk density of snow $\rho_s$, a proportionality applies to CDR (in dB) as follows (Ryzhkov and Zrnić, 2019):

$$\text{CDR} \approx \rho_s^2(D)(L_a - L_b)^2 = (\alpha_0 f_{rim} D^{-1})^2 (D)(L_a - L_b)^2, \tag{13}$$

where $\rho_s$ is expressed in g cm$^{-3}$, $L_{a,b}$ are the particle shape parameters, $\alpha_0$ is a constant that is approximately equal to 0.15, $f_{rim}$ denotes the degree of riming, which ranges from 1 for unrimed ice to 5 for heavily rimed ice and can be expressed as a function of RMF as $f_{rim} = 1/(1 - \text{RMF})$. CDR is mostly a function of shape, as the net effect of Eq. (13) is dominated by the more spherical shape of rimed snow compared to unrimed snow and not by the effect of increasing density, which ultimately results in a stronger reduction of CDR for rimed snow.

As a proxy of CDR, DR (in dB) can be estimated via

$$\text{DR} = 10\log_{10}\left[\frac{1 + Z_{dr} - 2\rho_{hv}Z_{dr}^{1/2}}{1 + Z_{dr} + 2\rho_{hv}Z_{dr}^{1/2}}\right], \tag{14}$$

where $Z_{dr}$ denotes the differential reflectivity in linear units. Eq. (14) combines the information content of $Z_{dr}$ and $\rho_{hv}$ in a single, more meaningful quantity. And due to the inherent noise reduction in the QVP methodology, an even clearer riming fingerprint can be expected. DR bears several advantages over CDR measurements and thus, is more robust. In contrast to CDR, DR does not depend on propagation phase shift, is available in all radar resolution volumes where directly measured co-polarized signals are reliably measured, and has a rather modest sensitivity to particle wobbling (Matrosov, 2020). In addition, unlike other polarimetric variables such as $Z_{DR}$, DR shows only weak dependence on the orientation of the hydrometeors and is less affected by noise (Ryzhkov et al., 2017). While low DR values are expected for almost spherical targets, high values indicate a wide variety of shapes or elongated targets. Aside from this shape dependence, the DR values also depend strongly on the particles density. Fig. 4 illustrates and quantifies the strong variability of DR with $\rho_{hv}$. In case of heavy riming, an increase of $\rho_{hv}$ is to be expected due to decreasing anisotropy, resulting in more negative DR values. Eventually, strong vertical DR columns in QVPs may help to better identify riming and its vertical extend aloft.

## 4.2 Workflow

This section describes the setup, training, and evaluation procedure of the tested algorithms and Fig. 5 summarizes the complete schematic workflow of the algorithm development. First, a preselection of relevant potential predictors based on manual feature engineering is done, i.e. taking the information content of the polarimetric variables into account. Therefore, the same set of polarimetric variables, namely $Z_H$, $Z_{DR}$ and DR, are fed as inputs to all approaches. $Z_H$ increases with increasing particle size and $Z_{DR}$ decreases with decreasing oblateness; both are well known effects of riming. DR was selected as it combines the information content of $Z_{DR}$ and $\rho_{hv}$ and suspected to potentially amplify the riming fingerprint (see Sect. 4.1).



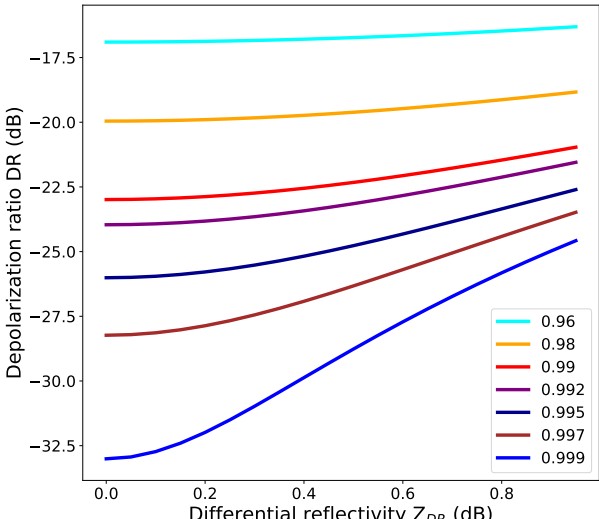

**Figure 4.** DR-$\rho_{hv}$ relations given by Eq. (14) for different values of $Z_{DR}$. The colors represent different values of $\rho_{hv}$.

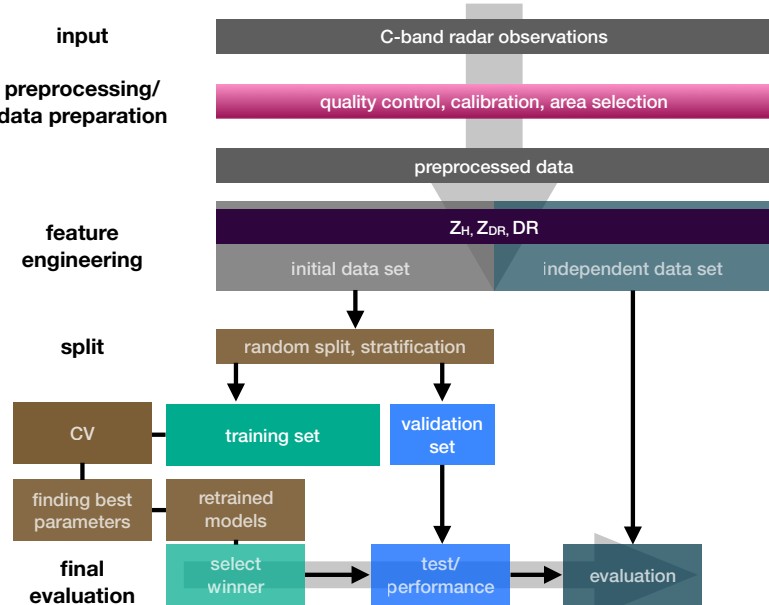

**Figure 5.** Graphical representation of the classification workflow for the algorithm development. The two steps performed after to cross-validation (CV), indicated as brown boxes, are only carried out for the GBM and the ANN but skipped for the other classifiers.

The overall data base is split in two parts, one used for the algorithm development and referred to as the initial data set and the other used for the evaluation and referred to as the independent data set (see again Table 1 and Fig. 5). The employed data



**Table 3.** Optimal hyperparameter values of tuned GBM and ANN.

| GBM hyperparameter | | ANN hyperparameter | |
|---|---|---|---|
| Learning rate | 0.15 | Learning rate | 0.001 |
| Maximum depth | 4 | No. of hidden layers | 1 |
| No. of boosting stages | 100 | No. of neurons | 8 |
| Subsample | 0.7 | Weight decay | 0.05 |
| Minimum sample split | 100 | Max. iteration | 1000 |

set for development is, however, just a subset of the initial data set and again split in a ratio of 70 % to 30 % into a training

and validation set, respectively (compare with Fig. 5). Both, the training set and the validation set, show the same proportions of riming and non-riming sequences like the initial data set (stratification), which is unbalanced (21 % were labeled as riming and 79 % as no-riming). The prediction task is to classify whether a riming threshold is exceeded by training all classifiers to predict Doppler velocities faster than 1.5 ms$^{-1}$ with a total of 16491 collocated data points within the initial data set. To optimize the performance of the different approaches, a cross-validation (CV) is performed on the training set to minimize

overfitting and to tune hyperparameters. Thus, this training data is again divided into k smaller sets (k-folds) of sub-training and sub-validation sets, whereby a split into five (k=5, five iterations) equally sized sets is chosen here. K-fold cross-validation is a method of validation frequently utilized in machine learning to assess the generalization ability of a prediction model. Model tuning is also performed to investigate the impact of hyperparameter configurations, that depend on the selected classifier, on the performances. The best possible set of hyperparameters from a pre-selected parameter space are found via a grid search

method. In summary, the training is performed on the effective training set, then the initial evaluation of all models is performed, and if the classification task appears to be successful, the evaluation of the winners of the (tuned) models can be performed on the remaining 30 % hold-out validation set. The last evaluation step is carried out on the independent data set for the best model found.

Table 3 shows the best performing set of identified model hyperparamters (tuning results) describing the structure for both the

GBM and the ANN as a reference for potential future applications. The latter uses a hidden layer with a hyperbolic tangent activation function.

Finally, we tested the TB using selected hard thresholds for all polarimetric variables included (DR <= -22.6 dB; 0.05 dBZ < $Z_{DR}$ < 0.21 dB; $Z_H$ > 10 dBZ) on the validation data set.

## 5  Results and Verification

First, the precipitation event observed on 2 January 2022 between 03:00 to 09:30 UTC is presented in more detail to illustrate the information content of the polarimetric variables, before in a second step, the methods described in Sect. 3.1 are applied to all events except the independent (Ahrtal flooding) case in Table 1 to set-up the riming detection algorithm.

Fig. 6 shows QVPs of $Z_H$, $Z_{DR}$, $\rho_{hv}$ and DR for the selected test case. Similar QVP products have been generated for all time



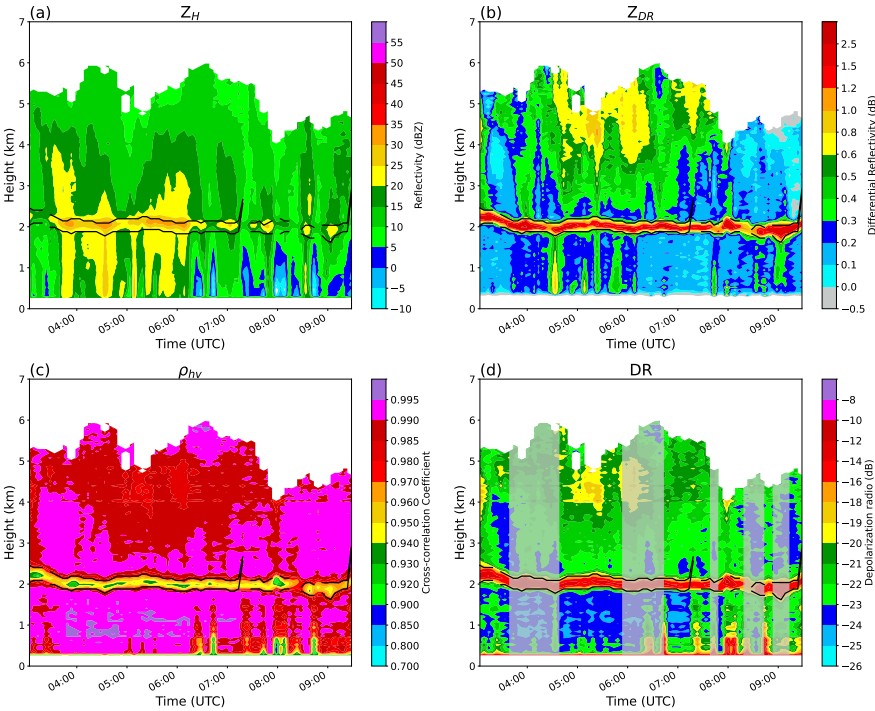

**Figure 6.** Composite QVP of $Z_H$ **(a)**, $Z_{DR}$ **(b)**, $\rho_{hv}$ **(c)**, and DR **(d)** on 2 January 2022 between 03:00 to 09:30 UTC. Profiles were constructed from the 12 degree elevation angle PPI scans. The time periods of DR identified as saggy periods are overlaid in transparent gray.

periods used (not shown here). ML signatures in terms of $Z_H$, $Z_{DR}$ and $\rho_{hv}$ are clearly visible during the entire period (Fig.

6). In addition, the QVP of DR [estimated from Eq. 14] in Fig. 6d indicates pronounced maxima within the ML approaching
-10 dB. Downward excursions or sagging of the ML (see also Kumjian et al., 2016 and Xie et al., 2016) may indicate riming
processes, however, also changes in precipitation intensity and associated cooling due to the enthalpy of melting may cause
these signatures (Carlin and Ryzhkov, 2019). To illustrate the connection between a sagging ML and riming, the signature is
detected by (1) applying a moving average to the ML top and bottom, respectively, (2) calculating the first derivative of both

time series, and (3) identifying the negative slopes of ML top and bottom. Indeed, the QVP of DR shows episodic sagging of
the ML mostly during time periods with noticeably reduced values of DR directly above the ML and up to altitudes of 4 km
(Fig. 6d).

 A correlation of 0.7 between MDV, derived from the MISPs of the corresponding radar birdbath scans and DR further empha-
sizes the strong potential of DR for riming detection (Fig. 7). The majority of the displayed data points are concentrated along

the one-to-one line, and the high correlation between $Z_{DR}$ and DR of 0.9 is obvious and not surprising, as DR is a function of
$Z_{DR}$ and $\rho_{hv}$. Nevertheless, DR alone is not sufficient to detect riming. Due to the dependence on $\rho_{hv}$, DR can exhibit quite
negative values together with relatively high $Z_{DR}$ values not expected in case of riming (see Fig. 4 and Fig. 7). It is also worth
mentioning, that due to the inherent averaging process in the QVP technique, potential DR values $< -28$ dB (see Fig. 4) are





not observed in our analyses.

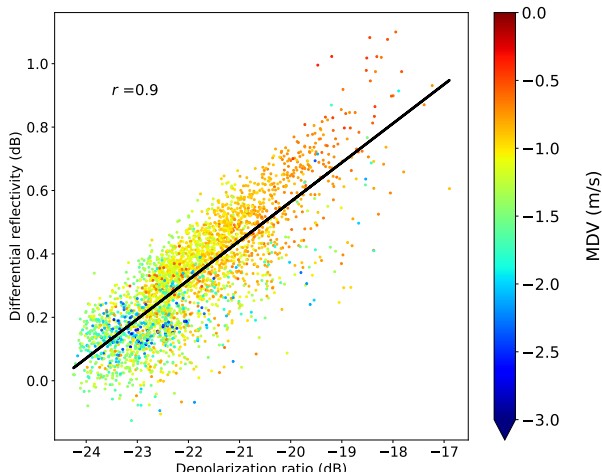

**Figure 7.** Scatterplot and linear regression of $Z_{DR}$ vs. DR observed with the DWD C-band radar in Essen on 2 January 2022 between 03:00 to 09:30 UTC. The coloring of the individual data point indicates MISPs of MDV. The correlation coefficient $r$ is provided for $Z_{DR}$ vs. DR.


**Table 4.** Comparative performance of TB, LR, QDA, GBM, ANN for classification of riming. The scores refer to the performance of the methods applied to the 30 % of the hold-out data set. The best scores of each metric are shown in bold.

| | **Performance measures** | | | | | | |
|---|---|---|---|---|---|---|---|
| **Classifier** | ACC | BA | F1 score | MCC | NMCC | J | $\kappa$ |
| TB | 0.77 | 0.58 | 0.32 | 0.2 | 0.6 | 0.19 | 0.19 |
| LR | 0.8 | 0.58 | 0.29 | 0.27 | 0.64 | 0.17 | 0.21 |
| QDA | 0.81 | 0.59 | 0.32 | 0.34 | 0.67 | 0.19 | 0.26 |
| GBM | **0.84** | **0.68** | **0.52** | **0.47** | **0.73** | **0.36** | **0.44** |
| ANN | 0.82 | 0.65 | 0.46 | 0.38 | 0.69 | 0.3 | 0.36 |

Secondly, based on these preliminary findings, a competition between all methods is performed for our selected test cases, i.e. the initial data set (see Table 1), monitored with DWD's ESS radar in order to identify the most appropriate algorithm for the discrimination task at hand. The extension to more test cases leads to a more robust, universally applicable algorithm that is less prone to potential minimal miscalibrations.

The evaluation of all methods as obtained with the training and validation procedure described in Sect. 4 is summarized in Table 4. The GBM-based riming retrieval outperforms all other classifiers, in terms of all performance measures. While the results of QDA and LR are comparable, the most simple TB performs slightly worse. Significantly better results than these are achieved by ANN, which has a performance close to that of the GBM. The results highlight that choosing a learning-





based method over a simple thresholding approach can significantly increase the prediction accuracy of riming. The inherent
impurity-based feature importance of the winning GBM retrieval, which indicates how effective each polarimetric input vari-
able for this specific model is, is composed of 43 % for $Z_H$, 30 % for $Z_{DR}$ and 27 % for DR. This decomposition states that
$Z_H$ and $Z_{DR}$ can already provide an first guess of where riming may occur. Despite the relatively minor contribution of DR,
it is crucial to localize and constrain these riming signatures with greater precision. This is because $Z_H$, for instance, tends to
classify artifacts that are discarded during the learning process when DR is incorporated. In addition, the three predictors are
not independent and may likely contain overlapping information. This is further investigated via Shapley values below.

In order to reduce the impact of possible mismatches, the GBM-based prediction is additionally smoothed in time and height via
a rolling minimum with window sizes of two. Figure 8 gives an impression of the performance of the final tuned and smoothed
GBM retrieval ($GBM_s$) applied to the complete initial input data set. Since the stratified test set is not a consecutive time series,
only the application to the initial data set allows to visualize the direct comparison of the predictions with the retrieval results
during the evolution of the riming processes. In all cases investigated the overall riming pattern is nicely represented and the
$GBM_s$ shows promising results with a balanced accuracy of 79 %, an F1 score of 0.61 and a NMCC of 0.78. Note that these
metric values are better compared to the ones obtained with the hold-out validation set (Table 4), which can be explained by
the smoothing and the fact the training data has been included. Additionally, the case of 3 November 2021 (initial data set)
demonstrates also the good performance of the $GBM_s$ algorithm when almost no riming is observed (Fig. 8b, f). This empha-
sizes that the GBM retrieval is also capable of correctly predicting the absence of riming, i.e. dominating aggregation processes.

It is also interesting to look at the polarimetric input variables associated with riming prediction and the mean degree of riming.
$GBM_s$ applied to the initial input data set results in a mean RMF of 0.47 with mean fall velocities of -1.66 ms$^{-1}$ and corre-
sponding mean values of DR = -22.67 dB, $Z_{DR}$ = 0.27 dB and $Z_H$ = 21.2 dBZ, respectively. Moreover, periods of a sagging
ML in Fig. 6d are consistent with both observed and predicted riming in Fig. 8c, d, adding another weight to the presence of
faster-falling particles above the ML.

To assess the performance of the final (smoothed) GBM retrieval and to investigate the transferability of the developed re-
trieval method to cases for which it has not been trained or validated, it is required to consider another independent data set.
Therefore, the retrieval and its smoothed variant are applied to a long-lasting intense stratiform precipitation event, which led
to devastating floods especially in western Germany in the Ahr valley in Rhineland-Palatinate on 14 July 2021 (e.g. Mohr et al.,
2022). The 17.5 h lasting event spanning from 01:00–18:30 UTC comprises a total number of 13050 collocated data points
whereas 20 % are labeled as riming and 80 % as no-riming. The predictions show convincing results (Table 5). The overall
performance of the GBM algorithm is better or equal compared to the performance of the GBM applied to the hold-out data
set, except for the metric of F1 score (0.52 vs. 0.51) and $J$ (0.36 vs. 0.34), for which nevertheless very similar values were
obtained. This underlines the robustness of the GBM retrieval. When comparing the metrics of GBM to $GBM_s$, the smoothed
variant performs only better in terms of BA (74 % vs. 78 %). In general the $GBM_s$ tends to overestimate riming occurrences,
but at the same time the amount of TP increases and the amount of FN decreases. This leads to the conclusion that smoothing,
which resulted in a better prediction of GBM applied to the initial complete data set, is not necessary for the independent data



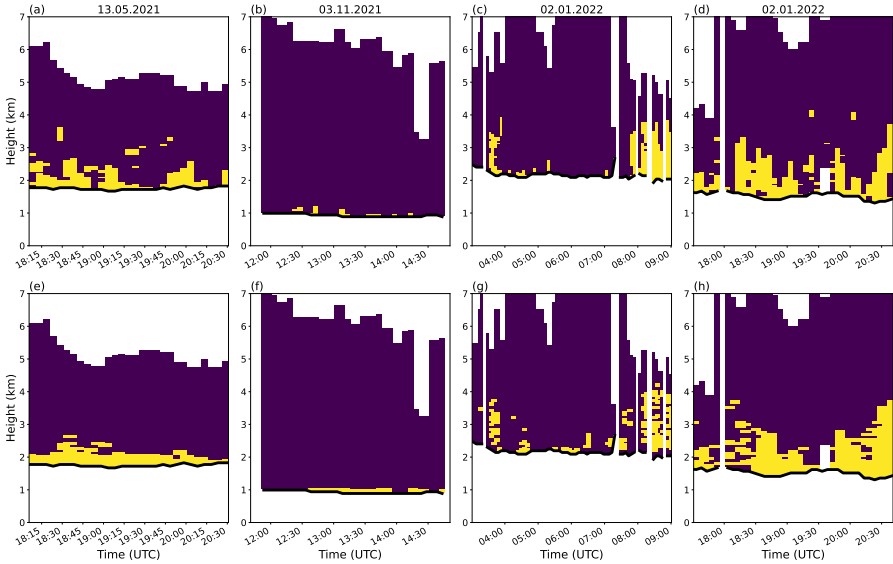

**Figure 8.** Binary time-height plots of MDVs faster than 1.5 ms$^{-1}$ (top panels, **a-d**) and corresponding GBM$_s$ retrieval results (bottom panels, **e-h**) for 13 May 2021 between 15:15 and 20:30 UTC (**a, e**), 03 November 2021 between 11:30 and 15:00 UTC (**b, f**), 02 January 2022 between 03:00 and 09:00 UTC (**c, g**) and 02 January 2022 between 17:00 and 21:00 UTC (**d, f**). (Predicted) riming is indicated in yellow, while no (predicted) riming is indicated in purple. ML tops are shown with black lines and the vertical stripes mark discarded data where no ML has been detected.

set, but also causes hardly any loss of performance. Both classifiers, GBM and GBM$_s$, reproduce the overall riming patterns nicely (Fig. 9).

For the Ahrtal flooding event, the GBM retrieval is able to detect riming with a mean RMF of 0.43. Following Kneifel and Moisseev (2020) this corresponds to a fall velocity of approximately -1.45 ms$^{-1}$ at C-band derived via the RMF-MDV polynomial fit in the Rayleigh regime. Moreover, the predicted data points exhibit mean values of DR = -21.21 dB, Z$_{DR}$ = 0.3 dB and Z$_H$ = 21.48 dBZ. These values for Z$_H$ and Z$_{DR}$ again correspond to the expected signatures for faster-falling particles, which tend to enhance the Z$_H$ and decrease the Z$_{DR}$ above the ML. The lower mean Z$_{DR}$ values are in line with riming signatures of particles that become more spherical resulting in a lower Z$_{DR}$ by 0.1–0.3 dB (Ryzhkov et al., 2016; Kumjian et al., 2016; Giangrande et al., 2016; Vogel et al., 2015). The mean MDV of all data points is -0.85 ms$^{-1}$, while the mean MDV for all points where riming is predicted is -1.51 ms$^{-1}$.

Even though the spatio-temporal mismatches caused by the comparison of a 15 s snapshot of the vertical atmospheric column with the average profile of a conical volume may lead to double penalties (Gilleland et al., 2009) affecting the scores, the developed GBM algorithm overall distinguishes reliably between dominant aggregation and intense riming processes.

As the GBM generally provides the most accurate results, the Shapley values $\psi$ are estimated for all unsmoothed GBM models and for each parameter combination. This is conducted on both the independent and the complete initial data set.





**Figure 9.** Binary time-height plots of MDVs faster than 1.5 ms$^{-1}$ (top) together with results of the GBM retrieval (middle) and the GBM$_s$ retrieval (bottom) for the flooding case on 14 July 2021 between 01:00 to 18:30 UTC monitored with the ESS radar. Colors and black lines like in Fig. 8.





**Table 5.** Metrics for the best performing GBM riming algorithm before and after smoothing (GBM$_s$). Scores refer to the performance applied to the independent data set. The best scores of each metric are highlighted in bold.

| | Performance measure | | | | | | |
|---|---|---|---|---|---|---|---|
| **Classifier** | ACC | BA | F1 score | MCC | NMCC | J | $\kappa$ |
| GBM | **0.94** | 0.74 | **0.51** | **0.47** | **0.74** | **0.34** | **0.47** |
| GBM$_s$ | 0.91 | **0.78** | 0.48 | 0.45 | 0.72 | 0.31 | 0.43 |

**Table 6.** Shapley values $\psi$ and their decomposition (listed in parenthesis in %) for the independent and initial data set. NMCC has been considered as metric for calculating the Shapley values.

| Predictor | Independent $\psi$ | Initial $\psi$ |
|---|---|---|
| Z$_H$ | 0.36 (49 %) | 0.31 (40.5 %) |
| Z$_{DR}$ | 0.18 (24 %) | 0.21 (28.5 %) |
| DR | 0.20 (27 %) | 0.24 (31 %) |

Thus, calculations are performed for a set of six different combinations for both data sets, whereby the GBM models using a
combination of two polarimetric variables or only one alone had to be retrained for their respective hyperparameters.

From the analyses of the Shapley values on the independent data set, Z$_H$ emerges as the top predictor with an importance of 49 %, followed by DR with 27 % (Table 6). DR has a higher impact than Z$_{DR}$ (24 %), but the difference is not significant. Other metrics, e.g. BA, also show a fairly balanced importance of DR and Z$_{DR}$ (not shown). Interestingly, considering the true negative rate (TNR), the Shapley values indicate that DR has the largest influence at 37 %. This may be due to the ability of
DR to limit the tendency of Z$_H$ to over-predict riming.

The Shapley values calculated for the initial data set show similar results as for the independent data set, also matching well with respect to the ranking of the predictors. However, a slightly higher importance of the variables DR (31 % vs. 27 %) and Z$_{DR}$ (28.5 % vs. 24 %) has been obtained. One potential reason for the greater impact of DR could be the presence of more intense and deep riming signatures in the initial data set, e.g. on 2 January 2022. Thus, learning these distinct riming features
from DR alone was possible, but not from Z$_H$ (not shown). Ultimately, the GBM retrieval using all three polarimetric variables outperforms the GBM models using just one or a combination of two on the independent data set. This indicates that the predictors employed improve each other.

In its final form, the novel algorithm performs well for all tested cases and is particularly encouraging in that in can be easily implemented in operational services for area-wide applications. This algorithm could be applied to identify dominant riming
conditions that pose a potential icing hazard to aviation. Although the final algorithm cannot be expressed as a simple equation because it is based on an ensemble of trees, it can be provided upon request as a stored ready-to-use model. In addition, the algorithm also enables the creation of a Germany-wide climatology of riming pre-2021, when no Doppler spectra were stored.





## 6 Conclusions

The overarching goal of this study was to develop an algorithm which enables the distinction between rimed and aggregated
snow based on polarimetric weather radar data alone. The introduced riming detection algorithm delivers promising results,
requires few computational resources and bears worldwide opportunities and advantages such as applicability to any slant-
viewing polarimetric weather surveillance radar, even those without vertical scanning strategy.

Tests with widely used binary scores identified the GBM algorithm as the one with the best performance. When considering
predictive BA for an independent case, the trained GBM riming retrieval was able to correctly predict about 74 % of observed
riming features and thus gives confidence to detect area-wide riming based on operational national radar networks. The un-
derlying assumption is that the polarimetric riming signature is dominating in the resolved radar volume and not obscured by
other ice particle types. Another challenge is that small-scale riming features may be obscured or reduced in magnitude due to
the inherent averaging procedure in the QVP technique.

The Shapley values highlight the to date underutilized DR estimator as a crucial contribution for the riming detection algo-
rithm. However, well calibrated $Z_{DR}$ is required as input as well as noise-corrected $\rho_{hv}$ to ensure the reliability of DR.

The algorithm was built on a limited number of training data sets and from the ESS radar only. In the future, a comprehensive
climatological training data set will enable to increase robustness and to further improve the performance of the GBM retrieval.
However, building such a data set is challenging, because the fully automated post-processing chain of the Doppler spectra can
still fail under extreme precipitation conditions and thus must be replaced by a simpler manual thresholding method, resulting
in increased computational time and cumbersome effort. Such an extended data base would also allow in the next logical step
the separation of fall speeds into several groups in order to learn multiple riming classes, such as moderate riming, heavy
riming, and potentially a graupel class characterized by fall velocities of up to 3 ms$^{-1}$. Similar to the envisioned different
riming classes in a future refinement of the algorithm, distinct classes for the aggregation process could also be introduced.
Towards this goal, the vertical gradient of $Z_H$ above the ML ($\beta = \partial Z_H / \partial z$) could be included as additional input variable to
the classifier.

The applicability of the algorithm to radars operating at different wavelengths, such as for the operational S-band radar net-
work of the National Weather Service (NWS) in the United States, also remains to be investigated in the future. Mechanical
constraints do not allow the implementation of a birdbath scanning routine in the scan schedule, as 20 degree is the largest el-
evation angle used (Matrosov, 2020) of the NWS network, requiring the use of alternative measurements as ground truth. The
ongoing extension of the NWS network with gap-filler radars operating at X- and C-band led by the private company ClimaVi-
sion in the United States, should allow for the direct application of our algorithm to at least their C-band radars. Deploying the
riming detection algorithm across radar networks in different climatic regions could potentially also prove beneficial in gaining
a deeper understanding of the importance of riming in the formation and evolution of precipitating clouds.

The radar signatures in QVPs, as used in this study, reveal the average dominating precipitation process within the monitored
radar domain (with a projected range of approximately 34 km for QVPs of 12 degree). A next potential extension would be
to explore the possibility to identify also smaller-scale riming conditions via columnar vertical profiles (Murphy et al., 2020),

process-oriented vertical profiles (Hu et al., 2023) or range height indicator sector vertical profiles (Blanke et al., 2023). Additional aircraft in-situ measurements of particle size distributions and the particle habits over the DWD C-band network domain would enable an even more detailed accuracy assessment and evaluation of potential applications of such extensions of the algorithm. Such coincident data could facilitate the development of an algorithm that also directly quantifies the degree of riming.

Finally, the riming detection algorithm is also of value for the evaluation of numerical weather prediction models and data assimilation. E.g., the benefit of state-of-the-art ice microphysical retrievals for these applications is currently investigated (e.g., Reimann et al., 2023; Trömel et al., 2021; Trömel et al., 2023), but most retrievals show reduced accuracy in the presence of riming. The novel riming detection algorithm could therefore be used to just mask those regions or even replace , in riming conditions, current retrievals by upcoming developments taking $f_{rim}$ into account (personal communication with Alexander V. Ryzhkov, 2024). Nonetheless, this study already illustrates the key components and capabilities of a solely radar-based riming detection algorithm, without any additional aids like vertically pointing devices.

*Code availability.* The code is available upon request by contacting the authors.

*Data availability.* To obtain DWD radar volume data and birdbath data including Doppler spectra, please contact DWD customer relations at kundenservice@dwd.de, following DWD regulations.

*Author contributions.* AB and ST jointly designed the study. AB led the writing, created the visualizations and carried out the analysis and data handling. ST helped in the analysis process. MG provided scientific expertise on Doppler spectra and contributed to their analysis. All authors contributed to the proof-reading of the final draft.

*Competing interests.* The contact author has declared that none of the authors has any competing interests.

*Acknowledgements.* The research was carried out in the framework of the special priority programme SPP-2115 "Polarimetric Radar Observations meet Atmospheric Modelling (PROM)" in the project POLICE. The C-band weather radar data from Essen radar was provided by the German national meteorological service (Deutscher Wetterdienst, DWD). We acknowledge the support of Kai Mühlbauer and the open-source radar library $\omega$radlib (https://docs.wradlib.org/en/2.0.0/, last access: 17 October 2024) regarding the processing of radar data.



485 *Financial support.* This research has been supported by the German Research Foundation (DFG, Deutsche Forschungsgemeinschaft; grant nos. TR 1023/13-1 and FR 4119/1-1). The article processing charges for this open-access publication were covered by the University of Bonn.



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
