# Peer review of "A new aggregation and riming discrimination algorithm based on polarimetric weather radars"

_EGUsphere, 2024_

## Author Comment (AC1)

Review of "A new aggregation and riming discrimination algorithm based on polarimetric weather radars", by Armin Blanke, Mathias Gergely, and Silke Trömel, egusphere-2024-3336.

**Response to reviewer 1**

Dear reviewer,
We are very grateful for your valuable feedback, which helped us to improve the manuscript. The manuscript has been thoroughly revised. Please find below our response, highlighted in blue. The revised manuscript is also provided with tracked-changes for better clarity.

General comments:

However, the methodology has notable limitations. The algorithm was developed using a limited number of events from a single radar station, which may constrain its generalizability to diverse meteorological settings. Its high sensitivity to input variables, particularly the accurate calibration of ZDR and ρhv, also poses challenges for widespread applicability, especially in operational radar networks outside Germany. These limitations should be explicitly acknowledged in the conclusion, rather than suggesting a direct application to radar networks like NEXRAD in the US without further validation and adaptation.

We thank the reviewer for these important comments and agree that we should be more precise in addressing the limitations mentioned above. It is true that an extension of the training data set by including more radar stations and diverse meteorological settings could enhance the generalizability of the model. This limitation has already been pointed out in lines 446f. However, we extended our previous formulation following your suggestion in order to emphasize it even more and explicitly: 'The algorithm was built on a limited number of training data sets and from the ESS radar only. In the future, a comprehensive climatological training data set that considers more radar stations and an even wider range of meteorological conditions will enable to increase robustness and to further improve the performance of the GBM retrieval'.
Indeed, the accuracy of $Z_{DR}$ and $\rho_{HV}$ is key for the algorithm, as stated in line 445 of the conclusion. Nonetheless, we are convinced that a sufficient calibration accuracy can be achieved, even for radar networks not having a birdbath scan on hand. In fact, this study exploited the $Z_H$ -$Z_{DR}$ relationship for C-band in light rain (Ryzhkov and Zrnić, 2019) instead of the birdbath scan due to the identified elevation dependency (see lines 104ff). The calibration based on the $Z_H$ -$Z_{DR}$ relationship or similar approaches can be achieved in other operational settings as well. Furthermore, we like to mention that despite the increasing use of phase-based measurements (specific attenuation A or specific differential phase $K_{DP}$), variables based on signal strength and thus affected by biases still provide valuable information and can not be completely avoided (see e.g. hybrid (ice) microphysical retrievals or rainfall retrievals when the radar monitors above the melting layer).
Similarly, we also agree with the referee that application to radar networks such as NEXRAD remains to be investigated and validated in order to assess to what extent an adaptation is needed. We included the need for further validation in our previous formulation in lines 456f: 'The applicability of the algorithm to radars operating at different wavelengths, such as for the operational S-band radar network of the National Weather Service (NWS) in the United States, also remains to be investigated and validated in the future.'

References:

Ryzhkov, Alexander V., and Dušan S. Zrnić. *Radar polarimetry for weather observations*. Vol. 486. Cham, Switzerland: Springer International Publishing, 2019.

---

## Author Comment (AC2)

Review of "A new aggregation and riming discrimination algorithm based on polarimetric weather radars", by Armin Blanke, Mathias Gergely, and Silke Trömel, egusphere-2024-3336.

**Response to reviewer 2**

Dear reviewer,
We are very grateful for your valuable feedback and suggestions to improve the manuscript. The manuscript has been thoroughly revised and point-by-point responses have been prepared. Please find below our replies highlighted in blue along with your suggestions. The revised manuscript is also provided with tracked-changes for clarity.

Specific comments:

1. Regions of significant riming used for training in this study are those with air density corrected values of the mean Doppler velocities greater than 1.5 m/s above the bright band (as observed using the vertical radar beam measurement geometry). These velocities approximate reflectivity-weighted snow/ice particle fall velocities which depend not only on degree of riming but also on the particle sizes. Larger particles (like those with higher Z values) would fall faster than smaller particles having the same riming degree (or rime fraction) but lower reflectivites (see, for example, eq.10.62 in Ryzhkov and Zrnic 2019). This is a reason for a positive correlation between MDV and reflectivity which is often observed. Decoupling the particle size and riming effects on MDV is challenging and a simple MDV threshold might not be universal. In any case, the authors need to address this issue in their discussions.

Thanks a lot for this valuable comment. Indeed, these challenges need to be emphasized in the paper, and it would become even more critical for a potential future refined algorithm distinguishing between light, moderate and heavy riming. In lines 151ff we already raised the point of using a fixed threshold: 'However, the potential occurrence of densely rimed dendrites, and lightly rimed aggregates, e.g. during the fill-in stage of riming growth (Heymsfield, 1982), is challenging to detect with such a fixed threshold value of 1.5 ms$^{-1}$. The fall velocity of such rimed particles may overlap with the ones of unrimed aggregates'. We now extended these explanations according your input: 'Furthermore, larger particles/aggregates, mostly associated with higher $Z_H$ values, fall faster than smaller particles with the same riming degree (or fraction of riming) but lower reflectivities (e.g. Ryzhkov and Zrinic 2019).'
Additionally, we now come back to this point in Section 5 lines 472ff and add the sentence: 'As already outlined in Sect. 2.2, larger particles would fall faster than smaller particles with the same degree of riming, and thus, it is challenging to decouple the effects of particle size and riming on MDV. Therefore, a strict MDV threshold may not be universal, but again, an in situ data base would help and allow for a decoupled approach.' Nevertheless, we are confident that the threshold used is appropriate for training the distinction between dominant rimed particles and aggregates even with some overlap in fall velocities.

2. The 15-sec temporal averaging of vertical beam data to minimize vertical air motions influence on Doppler velocity measurements may be insufficient (line 145). More quantitative justification of such averaging is needed.

Thank you for raising this point. Due to the scanning strategy we are limited to the 15 second averaging. In contrast to cloud radars, which measure spectra in continuous operation and thus allow uninterrupted averaging of successive time steps, the DWD's operational scanning strategy involves entire volume scans between the measurements of the spectra. Still, we agree that an even larger averaging period would further reduce the influences on the vertical air motions. One alternative would be to average over several scan cycles, e.g. over three measured spectra of about 15 minutes, which would then however lead to a rather coarse temporal resolution. Depending on the application, an approach with greater averaging could be suitable such as for quantifying the impact of gravity waves (e.g. Teisseire et al., 2024).

3. The authors suggest that riming leads a decrease in Zdr (line 273). It may not be so for initial stages of riming when supercooled water freezes and is deposited in between the crystal arms, so the bulk density increases but the overall shape remains approximately the same.

Thank you for this suggestion. We agree with the reviewer that for initial stages of riming the shape remains constant. We described the corresponding fill-in stage of riming growth describing in lines 151ff in the context of fall speeds: 'However, the potential occurrence of densely rimed dendrites, and lightly rimed aggregates, e.g. during the fill-in stage of riming growth (Heymsfield, 1982), is challenging to detect with such a fixed threshold value of 1.5 ms−1.' However, the decrease in $Z_{DR}$ mentioned in line 273 refers to heavy riming. We have reformulated the sentence in line 274 of the revised manuscript for clarification: 'On the one hand, the ice particles become more spherical due to riming, which leads to a reduction in $Z_{DR}$ beyond the initial fill-in stage of riming growth (Sect. 2.2)'.

4. As I understand Fig. 4 show theoretical calculations of DR. The authors also mention that due to QVP averaging DR values were > -28 dB (line 348). What minimal values of DR were obtained for instantaneous measurements with the slanted and vertical beam geometries?

Reliable minimum values of around -32 dB were obtained on the basis of the PPI measured at 12 degree elevation. However, before calculating QVPs, the distribution of DR values shows that the majority of values are greater than -28 dB. Due to the lack of information content of $Z_{DR}$ in the vertical, it is not possible to calculate meaningful DR values.

5. Did you account for the elevation angle changes of polarimetric variables when constructing QVP from slanted beam measurements at different elevation angles? Are the polarimetric variables used in this study recalculated for the horizontal beam pointing?

No, we did not consider changes in elevation angle as we only developed and used the algorithm for an elevation of 12 degrees. We are referring to the rather small impact around these elevation angles, as shown for example in Fig. 12 of Ryzhkov et al. (2005) for $Z_{DR}$. In snow, there is a neglectable dependence on the elevation angle between 12 degrees and the horizontal.

6. There are some features seen in Fig.6 which need explanations. For example, similar columns of rain reflectivities (~25 dBZ) just prior 4:00 and around 4:40 correspond to very different Zdr values (0.8 and 0.3 dB). High DR values in the region of supposedly drizzle (Z there is less than 0 dBZ) at around 8:00 UTC (Height < 0.5 km) also look strange. Are those artifacts of the QVP approach? By the way, adding the temperature profile to Fig. 6 would be useful.

We thank the reviewer for these important comments. Yes, the rain columns appear to look similar in Fig. 6 due to a combination of the QVP approach and the colorbar used. In the figure below, the same QVP for $Z_H$ is shown with different colorbar ticks. The earlier column around 4:00 UTC has reflectivities of 21 dBZ with maxima close to the ground and ML of 23 dBZ, while the column around 4:40 UTC is more intense with reflectivities of 24 dBZ up to almost 29 dBZ, which explains at least part of the corresponding differences in $Z_{DR}$.

[Figure]

Figure A: QVP of $Z_H$ of the Essen radar (ESS) at 12 degrees elevation on 2 January 2022.

As DR is calculated from $\rho_{hv}$ and is very sensitive to this variable, the high DR values are due to the use of this proxy. It is therefore more likely that this is an artifact. However, all of these artifacts below the melting layer are not included in our method as we are focusing on the ice phase only.
We appreciate your suggestion and have added -5, -10 and -15°C isotherms from ERA5 (Hersbach et al., 2020). The isotherms indicate that the pronounced DR columns are mostly located at temperatures above -10°C, where riming is more favoured (Kneifel and Moisseev, 2020). We replaced Fig. 6 with the revised version containing isotherms and added relevant information in the figure caption and in the following text of the revised manuscript:

Caption Fig. 6: 'Overlaid dashed lines (in all panels) display the -5, -10 and -15°C isotherms from the European Centre for Medium-Range Weather Forecasts Reanalysis v5 (ERA5; Hersbach et al., 2020).'
Line 339: 'A band of enhanced $Z_{DR}$ values is visible within the DGL located between -10 and -15°C (Fig. 6d).'
Lines 346f: 'These pronounced DR columns are mostly located at temperatures above -10°C, where riming is more favoured (Kneifel and Moisseev, 2020).'

7. There are very high MDVs (~ 2.5 m/s) above the bright band after 8:00 UTC (Fig.3). What kind of riming can be expected there? There could be contributions from vertical air motions. Doppler spectra could provide additional information.

Indeed, vertical air motions may have a significant impact on the Doppler spectrum. However, the example of an isolated Doppler spectrum measured on 2 January 2022 at 08:20 UTC displayed below shows a fairly regular spectrum. It clearly differs from the characteristic zigzag shape expected for extreme precipitation (Gergely et al. 2022). In our case, there is no broad precipitation mode, which indicates a minor influence of turbulent mixing of precipitation particles from different altitudes. Additionally, a melting layer is present, which is not the case for strong vertical air motions or up- and downdrafts caused by convection. We conclude, that the high MDVs are associated with intense riming.

[Figure]

Figure B: Isolated average Doppler spectra after postprocessing of a birdbath scan recorded on 2 January 2022 at 08:20 UTC.

8. I wonder why you did not consider any cold cases without rain and melting layer. Without intervening rain and ML, you could use microwave radiometer measurements of supercooled LWP to better identify riming conditions.
Therfore,  a simple MDV threshold may not be universal, but  again, an in situ data base would help and allow for a decoupled approach

Good point. In Germany, we generally observe very few pure snow events, especially for radar stations at lower altitudes above m.s.l, such as the Essen radar. This specific site is used since it is the only directly collocated radar with a sounding station that is available for the DWD network.
In addition, no directly collocated data from a microwave radiometer is available for this radar. The closest option is the 14-channel microwave profiler Humidity and Temperature Profiler (HATPRO; radiometer Physics GmbH, Germany; Rose et al. 2005) located at Forschungszentrum Jülich in western Germany (approximately 75 km distance to the Essen radar), which is too far away for our method. We agree and believe that additional collocated measurements of supercooled liquid water path could provide further insights into the riming conditions, if available.

9. Assuming that particles at cloud tops (~ 7 km) are small enough to be tracers, one can conclude from Fig.3 that vertical air motions of an order of 0.5 m/s could be present. Air motions of such magnitude could also be expected at lower heights. This would affect the identification of riming using the MDV threshold.

Since we are using a longer wavelength polarimetric radar operating at C-band wavelength, we cannot assume that the particles at the cloud tops are small enough to serve as tracers, as the radar used has insufficient sensitivity to detect echoes from small cloud particles less than 50-100 microns in size (see, e.g. Ryzhkov et al. 2020). However, we agree that there might be remaining uncertainties due to vertical air motions, turbulence and gravity waves, which are not fully removed by the averaging approach used.
The potential effect of extending the averaging period on the performance of the algorithm could be a subject for future research. Nevertheless, even the use of an averaging window, such as that introduced for instance in Mosimann (1995), may not be sufficient to cancel out moderate vertical air movements due to updrafts and downdrafts. Finally, we only use cases with a distinct melting layer where we can ensure that stratiform conditions do not seriously disturb the MDVs.

10. Define D in equation (13). I believe that this equation is written assuming the Rayleigh scattering approximation for spheroidal particles with vertical symmetry axes as viewed with the horizontal radar beam. It also assumes that the particle bulk density is proportional to the reciprocal of particle size. Also, I am not sure about the multiplication factor (D) in the middle of the right-hand side of this equation. Is it a typo? Finally, I believe that this equation is written for CDR in liner units not in logarithmic units of dB as stated in line 280.

Thanks for pointing this out! Yes, the equation is written assuming the Rayleigh scattering approximation and assuming that the bulk density of Rayleigh scatterers is inversely proportional to the particle size, i.e. equivolume diameter D. Indeed, this is a typo. We removed the factor (D) from the right hand side of the equation. The revised equation in dB units is as follows:

$$CDR \approx 10\log_{10}[\rho_s^2(D)(L_a - L_b)^2] = 10\log_{10}[(\alpha_0 f_{rim}D^{-1})^2(L_a - L_b)^2]$$

Furthermore, the aforementioned assumptions are now included in the revised manuscript in lines 282ff: 'In the Rayleigh scattering regime, the following proportionality applies to CDR (in dB) assuming dry aggregated snow with low bulk density $\rho s$ inversely proportional to the equivolume diameter D (Ryzhkov and Zrnić, 2019): [...]'

References:

Gergely, Mathias, et al. "Doppler spectra from DWD's operational C-band radar birdbath scan: sampling strategy, spectral postprocessing, and multimodal analysis for the retrieval of precipitation processes." *Atmospheric Measurement Techniques* 15.24 (2022): 7315-7335.

Hersbach, Hans, et al. "The ERA5 global reanalysis." *Quarterly Journal of the Royal Meteorological Society* 146.730 (2020): 1999-2049.

Kneifel, Stefan, and Dmitri Moisseev. "Long-term statistics of riming in nonconvective clouds derived from ground-based Doppler cloud radar observations." *Journal of the Atmospheric Sciences* 77.10 (2020): 3495-3508.

Mosimann, L. "An improved method for determining the degree of snow crystal riming by vertical Doppler radar." *Atmospheric research* 37.4 (1995): 305-323.

Rose, Thomas, et al. "A network suitable microwave radiometer for operational monitoring of the cloudy atmosphere." *Atmospheric research* 75.3 (2005): 183-200.

Ryzhkov, Alexander V., et al. "Calibration issues of dual-polarization radar measurements." *Journal of Atmospheric and Oceanic Technology* 22.8 (2005): 1138-1155.

Ryzhkov, Alexander V., et al. "What polarimetric weather radars offer to cloud modelers: Forward radar operators and microphysical/thermodynamic retrievals." *Atmosphere* 11.4 (2020): 362.

Teisseire, Audrey, et al. "Attribution of riming and aggregation processes by application of the vertical distribution of particle shape (VDPS) and spectral retrieval techniques to cloud radar observations." *EGUsphere* 2024 (2024): 1-28.